# Application of Micron-Sized Zero-Valent Iron (ZVI) for Decomposition of Industrial Amaranth Dyes

**DOI:** 10.3390/ma16041523

**Published:** 2023-02-11

**Authors:** Dominika Ścieżyńska, Dominika Bury, Michał Jakubczak, Jan Bogacki, Agnieszka Jastrzębska, Piotr Marcinowski

**Affiliations:** 1Faculty of Building Services, Hydro, and Environmental Engineering, Warsaw University of Technology, Nowowiejska 20, 00-653 Warsaw, Poland; 2Faculty of Materials Science and Engineering, Warsaw University of Technology, Wołoska 141, 02-507 Warsaw, Poland

**Keywords:** zero valent iron, dyes, oxidation processes, acid amaranth

## Abstract

Dyes are highly toxic and persistent in the environment. Their presence in water causes environmental and social problems. Dyes must be effectively removed from the water. A UV/ZVI/H_2_O_2_ process was applied to decompose two organic dyes, AM E123 and AM ACID. A commercial ZVI product, Ferox Flow, was used, and its properties were determined using SEM and XRF. The zeta potential, surface area, and optical properties of ZVI were also determined. The efficiency of dye removal in optimal conditions was 85.5% and 80.85% for AM E123 and AM ACID, respectively. Complete decolorization was observed in all samples. The decomposition of both dyes occurred according to a modified pseudo-second-order reaction and there was a statistically significant correlation between the TOC decrease, pH, and process time. The catalyst was observed to have high stability, and this was not affected by the performance of the treatment process even after the third cycle, as confirmed by the results of the catalyst surface analysis and iron diffusion test. Slight differences in process efficiency were observed after each cycle. The need for only a small amount of catalyst to decompose AM E123 and AM ACID, coupled with the ability to reuse the catalyst without the need for prior preparation, may reduce catalyst purchase costs.

## 1. Introduction

Unprecedented industrial development has been observed in the last few decades. As many new technologies and products appear, new chemical compounds are created. Unfortunately, manufacturing usually involves the generation of wastewater. Furthermore, cosmetics, textiles, food products, and many other industries widely use various substances in their production processes and many of these substances are highly toxic and persistent in the environment. Therefore, the generated wastewater requires appropriate treatment in order to prevent the leakage of these toxic chemical compounds into the environment [1].

Generally, even a high pollution load in illegally discharged, untreated wastewater looks similar to fresh water. The exceptions are immediately visible dye contaminations, even in minimal quantities. Notably, dyes are one of the substrates most often used in industrial production [2] and the color of the water in the reservoir into which wastewater with dye is discharged arouses the local community’s attention.

The rapidity with which new synthetic color compounds with various properties are created, the adverse, often harmful effects of their release into the environment, and the increased interest of the public, have forced the industry to adopt continuous monitoring, and adapt treatment methods accordingly.

The most common wastewater treatment technologies are physical, chemical, and biological treatments [3]. Using any of these methods to remove pollutants can be highly efficient [4]. Unfortunately, limitations such as high cost [5], formation of byproducts [6], high efficiency only under very restrictive conditions, or low applicability often reduce their attractiveness. Therefore, multistep processes utilizing several technologies are usually required to achieve satisfactory results. Consequently, there is a need to identify new, effective, and relatively inexpensive methods for removing both toxic components and color from dye wastewater.

Advanced Oxidation Processes (AOPs) are some of the most popular non-biodegradable industrial wastewater treatment methods. AOPs are based on generating strong oxidative radicals, which then oxidize organic compounds [7].

Theoretically, radical oxidation could completely oxidize compounds to carbon dioxide and water. In practice, a large variety of smaller molecular mass organic compounds are created due to a chemical reaction. The beneficial reaction products, obtained from the presence of oxygen-containing functional groups, are usually more polar, possess more extensive solubility in water, and are less toxic than the parent compounds.

The first described AOP, the modification of which is also the subject of this article, was the Fenton process, which is based on the generation of radicals via aqueous iron ions with hydrogen peroxide reactions. Fundamental to the process are two main reactions. The Fenton reaction (1) describes the Fe^2+^ to Fe^3+^ oxidation, with H_2_O_2_ converting to HO^●^. The second, slower reaction (2), also known as the pseudo-Fenton reaction, allows catalyst, Fe^2+^, recovery, and secondary radical production.
Fe^2+^ + H_2_O_2_ → Fe^3+^ + HO^−^ + HO^●^(1)
Fe^3+^ + H_2_O_2_ → Fe^2+^ + HO_2_^●^ + H^+^(2)

The Fenton process consists of two treatment mechanisms. Firstly, radical oxidation in acidic conditions, coagulation, and precipitation in alkalic conditions. Because of process limitations, including a need for many reactants and the generation of a high volume of after-process sludge, many researchers have focused on its modification and alternative sources of Fe^2+^ ions have begun to attract attention. It was found that a partial solution to this problem is the application of heterocatalysts such as nanoparticles [8], magnetic resins [9], and porous composites [10]. Using a solid catalyst in the process introduces heterogeneous reactions that occur at the phase separation boundary [11]. The grinding and texture of the catalyst increase the active surface area in the process. Typically, solid catalysts can also be recycled and regenerated relatively easily.

The zero-valent iron (ZVI or Fe^0^) catalyst was developed to modify the Fenton process. Fe^0^ oxidation is an alternative to the Fe^2+^ introduction method in this modification. The heterogeneous Fenton process also involves many reactions related to the catalyst surface (3)–(10).
Fe^0^ + O_2_ + 2H^+^ → Fe^2+^ + H_2_O_2_(3)
Fe^0^ + 2H^+^ → Fe^2+^ + H_2_(4)
2Fe^3+^ + Fe^0^ → 3Fe^2+^
(5)
Fe^0^ + H_2_O_2_ + 2H^+^ → Fe^2+^ + 2H_2_O(6)
Fe^0^ → Fe^2+^ + 2e^−^(7)
2Fe^0^ + O_2_ + 4H^+^ → 2Fe^2+^ + 2H_2_O(8)
Fe^0^ + 2H_2_O → Fe^2+^ + H_2_ + 2HO^−^(9)
4Fe^0^ + 3O_2_ + 6H_2_O → 4Fe^2+^ + 12HO^−^(10)

The ZVI/H_2_O_2_ process has been successfully used to remove pollutants from many types of wastewater, including pharmaceutical [12,13], cosmetic [14], flue gas desulfurization [15], hydraulic fracturing flow back fluid [16], landfill leachates [17], natural organic matter [18], and textile manufacturing plant [19]. Furthermore, the use of ZVI to remove dyes of different specifications—belonging to different dye classes, e.g., xanthene (rhodamine B) [20], or azo dyes such as Orange II [21], Congo red [22], and thiazine (methylene blue) [23]—is gaining popularity.

An additional element that increases the efficiency of the Fenton process and its modification is exposure to ultraviolet (UV) radiation. Photon-induced photocatalysis accelerates the catalytic reaction. A valence electron is knocked out of the catalyst surface in the heterogeneous process using UV radiation. The knocked-out electron and the resulting electron hole in the catalyst are the initiating agents of a series of chemical reactions, resulting in the formation of superoxide ions (O_2_^−^) and hydroxyl radicals (HO^●^).

In this study, both of the modifications mentioned above, heterogeneous catalyst and simultaneously used UV irradiation, were used to increase process effectiveness in the decomposition of amaranth dyes used by industry as a substrate. Most research on dye removal using catalytic methods is focused on using nanomaterials, which usually have superior properties but are generally very expensive, excluding them from practical, industrial applications. This research aimed to determine whether an inexpensive commercial product containing micron-sized pure ZVI as a heterocatalyst can be effectively implemented to remove selected dyes. The removal of dyes was controlled based on the removal of color and total organic carbon (TOC). The properties of the ZVI were determined both before and after the treatment process using Scanning Electron Microscopy (SEM), Fourier Transform Infrared Spectroscopy (FTIR), and X-Ray Fluorescence (XRF). Zeta potential, size characterization, circularity, surface area, porosity, and direct band gap as well as the stability of the catalyst and the possibility of its reuse were also determined.

## 2. Materials and Methods

### 2.1. Dyes

This work used ZVI as a catalyst in a modified Fenton process to remove two industrially used dyes. The first dye, Amaranth E123 (AM E123, synonyms: Acid Red 27, Azorubine S, Food Red 9; Food Colours Perczak Sp. J., Piotrków Trybunalski, Poland), is 100% Trisodium 2-hydroxy-1-(4-sulfonato-1-naphthylazo)naphthalene-3,6-disulfonate with empirical formula C_20_H_11_N_2_Na_3_O_10_S_3_. It is commonly used as a food coloring agent and cosmetic dye, as well as in other industries [24]. Exposure to the dye may result in irritation of the eyes, skin, and respiratory system. The second dye, Acid Amaranth (AM ACID; Boruta-Zachem Kolor S.A., Bydgoszcz, Poland), is a mixture of monoazo dyes, identified as Acid Red 27 with different additives which contain fillers and application enhancers. This dye has industrial applications and is mainly used in the textile industry for dyeing protein fibers (such as wool and natural silk), wood, leather, and household chemicals. Exposure may lead to respiratory system and eye irritation, mainly by mechanical means. Furthermore, temporary corneal staining is possible. Both dyes are utilized and provided by a cosmetic factory located in central Poland, which uses them as reagents in the production of cosmetics.

### 2.2. Zero Valent Iron

The ZVI, Ferox Flow, was supplied in powder form by Hepure (Hillsborough, NJ, USA).

The characteristics, in particular the surface and morphology of the powder, were analyzed at an accelerating voltage of 2.0 kV using a scanning electron microscope (SEM, Hitachi S3500N) after placing the ZVI on carbon tape and gold sputtering using BAL-TEC SCD 005. The digital photo of samples was prepared by a Leica DMS1000 microscope (Leica Microsystems GmbH, Wetzlar, Germany).

The surface area and porosity of the ZVI were examined via physical nitrogen sorption using a Quadrasorb-SI (Quantachrome Instruments, Germany) and recorded using a Flo Vac degasser (Anton Paar GmbH, Graz, Austria). To accurately analyze the structure of the material and the specific surface area (S_BET_), the Brunauer–Emmet–Teller method (BET) was used. In addition, the total pore volume (V_pore_) and average pore size (D_pore_) were defined using the Barret–Joyner–Halenda (BJH) method. The adsorption and desorption processes were prepared in a liquid nitrogen bath at −195 °C.

The ZVI composition was analyzed using X-ray fluorescence (XRF, PI 100, Polon-Izot, Warsaw, Poland) with rhodium (Rh) anode. The study was performed using a silicon drift detector (SSD), 125–140 eV resolution, and a multilayer monochromator of 50 keV. Studies were performed for powder samples and in an air atmosphere using a measurement time of 300 s per sample.

Optical properties, such as the absorbance, transmittance, and diffuse reflectance of the ZVI, were studied using double-beam scanning on a UV-Visible Spectrophotometer (Evolution 220, ThermoFisher Scientific, Waltham, MA, USA). For measurement, the integrating sphere was utilized. The analysis was performed at a wavelength range from 220 to 1100 nm. The following parameters were selected for the study: an integration time of 0.3 s, a resolution of 1 nm, and a scanning speed of 200 nm min^−1^. Based on the results obtained, the indirect band gap was evaluated using Tauc’s plot method (11).
(αhν)^r^ = B(hν − E_g_) [(eV cm^−1^)^2^](11)
where “α” is the absorption coefficient, “B” and “E_g_” are the band tailing and the band gap energy parameters, respectively, and “r” = 2.

### 2.3. Process Influence on Catalyst Morphology and Structure

The Zeta potential showed the stability of the material in the dispersion of the ZVI particles. The analysis was carried out using a DTS1060 cell with the NANO ZS ZEN3500 analyzer (Malvern Instruments, Malvern, UK), with the back-scattered light detector at an angle of 173° at 25 °C.

The parameters of the ZVI, in particular shape and size, were analyzed using a dynamic image analyzer (Sentinel Pro, Micromeritics Inst. Corp., Norcross, GA, USA). The device was equipped with a peristaltic pump and stroboscope camera. The equivalent circular area diameter model was chosen for sample characterization and focused on diameter (ECAD) and circularity.

The characteristics of the organic and inorganic groups on the ZVI surface were measured using Fourier transform infrared spectroscopy (FTIR, Nicolet iS5 FTIR Spectrometer, Thermo Scientific, Waltham, MA, USA) with attenuated total reflectance (ATR). The analyzer was equipped with a diamond crystal.

### 2.4. Dye Decomposition Using UV/ZVI/H_2_O_2_

Decompositions of amaranth dyes were performed in 1.5 L glass reactors with 1 L samples and the relevant amounts of individual reactants placed directly under an ultraviolet (UV) radiation source. The UV light irradiation was provided by medium pressure Fe/Co 400 W lamps of the type HPA 400/30 SDC, with 94 W UVA power (Philips, Amsterdam, The Netherlands). The samples were stirred at 300 rpm on magnetic stirrers (Heidolph MR3000, Schwabach, Germany) to prevent catalyst aggregation. The processes were carried out at a strictly controlled pH. The pH was increased to 9.0 using 3 M NaOH (POCh, Gliwice, Poland) to terminate the reaction after selected process times of 5, 10, 15, 30, and 60 min. Next, the post-process solutions were left for 24 h for sedimentation of iron hydroxide and decomposition of unreacted H_2_O_2_ (Stanlab, Lublin, Poland).

Additionally, the contribution of adsorption, the photolysis process, and the process using only H_2_O_2,_ and the catalyst were verified to analyze the influence of their individual contributions to dye decomposition.

The total organic carbon (TOC) was used to study the decomposition of the dyes. The measurements were performed following the EN 1484:1999 standard using a TOC-L analyzer (Shimadzu, Kyoto, Japan) equipped with an OCT-L8-port sampler (Shimadzu, Kyoto, Japan).

The dye decomposition kinetics concerning the TOC values were determined. ANOVA with a 0.05 significance level was used to study the magnitude of variability in the average concentrations of TOC.

In addition, absorbance changes in the 530 nm wavelength were monitored using a UV-Vis Hach DR 6000 spectrophotometer (Hach, Ames, IA, USA).

Finally, the potential to reuse the materials without regeneration after the process was investigated. After the first process, the samples were separated magnetically and used as a catalyst in a new treatment cycle with fresh amaranth dyes and UV-light irradiation.

### 2.5. Reactive Oxygen Species Study

The presence of free hydroxyl radicals was analyzed using a commercial indicator of a free radical sensor. The study was based on the activity of singlet oxygen levels utilizing Singlet Oxygen Sensor Green fluorescent reagent (Thermo Fisher Scientific, Waltham, MA, USA). Firstly, 250 µL of the dye was added to the multiwall plate. In the next step, the dissolved fluorescent reagent was transferred to the prepared sample to a final concentration of 5.5 µM. The sample was then incubated in the dark for 60 min. Finally, the fluorescence intensity was measured using a microplate reader (Infinite 200 PRO, Tecan, Männedorf, Switzerland) on 485 and 520 nm excitation and emission, respectively.

### 2.6. Iron Diffusion Test

Material stability investigations were carried out via a diffusion test. Bacteria and yeast from the American Type Culture Collection (ATCC, Manassas, Va, USA) were selected for this study. The diffusion tests were conducted with *Escherichia coli* (ATCC 10799), *Staphylococcus aureus* (ATCC 29213), and *Bacillus subtilis* (ATCC 11774), as well as representative yeast *Candida albicans* (ATCC 14053).

Firstly, fresh, 24-h suspensions (0.5 McFarland) of the abovementioned microorganisms in phosphate-buffered saline (PBS) were prepared for the performance of the diffusion tests. Next, Petri plates with a solid agar medium (Merck, Rahway, NJ, USA) were inoculated using a sterile cotton swab. The inoculation was left to dry for 10 min and then the tested materials (c.a. 30 mg) were transferred with a sterile spatula. Importantly, the samples were placed next to each other at a distance of at least 24 mm and 10 to 15 mm from the edge of the Petri plate to ensure that the diffusion zones around the samples did not overlap. Finally, the Petri plates were incubated for 24 h at 37 °C. After incubation, digital images of Petri plates were obtained, and the diffusion zones were measured. Ten subsequent measurements were performed using ImageJ software (National Institutes of Health and the Laboratory for Optical and Computational Instrumentation, USA) to assess the final result. Additionally, the standard deviation (SD) was calculated to estimate the measurement error.

## 3. Results

### 3.1. ZVI Characterization

Material characteristics, such as a large surface area, porosity, and grain sizes, are essential in selecting the most effective material for wastewater treatment, and its chemical composition and optical properties are no less important. Therefore, the first step was to characterize the ZVI.

Firstly, the most important material parameters, such as morphology and structure, were verified. The ZVI was utilized in the form of a powder containing small grains with a characteristic metallic sheen (Figure 1a). The material had an irregular microstructure with an elongated shape. Grain sizes were smaller than 250 μm, which can be seen in Figure 1b. The iron surface was characterized by a smooth structure with recesses, which increased the specific surface of the material (Figure 1c,d). No additional compounds that could change the properties of the iron were observed.

Next, the optical properties of ZVI, such as the absorbance (Figure 1e) and direct bandgap (Figure 1f), were analyzed. The results showed excellent light absorption from 300 to 1000 nm. These perfect properties could be attributed to the black color of the material. The ZVI grains had an optical absorption edge of approximately 300 nm. This effect is generated by the intrinsic absorption band involved in the bandgap transition. Thus, the results suggested the presence of transmission bandgaps, which were then evaluated using Tauc’s plot method. A direct bandgap with a value of 2.2 eV was calculated. This parameter was similar to other types of justified iron, especially hematite and magnetite (2.2 eV), and goethite (2.1 eV) [25,26]. The excellent optical properties of ZVI indicated that the material was suitable for the photo-based process.

In the next step, specific surface area and porosity studies were performed using the physical nitrogen sorption method. The results obtained showed a surface area with a value of 1.711 m^2^ g^−1^, smaller than that of other iron materials, including hematite [27], goethite [28], or magnetite [29] (Appendix A). In addition, the average pore size of the material value was 3.20 nm, and the mean pore radius was 2.0 nm, which suggests that the material had excellent adsorption properties (Appendix A). The physical properties of the ZVI were obtained using the physical nitrogen sorption method using the BJH (Barrett–Joyner–Halenda) method and the single-point BET model. The results revealed that the surface area S_BET_ had a value of 1.711 m^2^ g^−1^. The average pore surface S_pore_ was 1.507 m^2^ g^−1^ pore volume, V_pore_ 0.003 cm^3^ g^−1^, and the pore diameter D_pore_ was 20.201 nm. The S_BET_ value was practically equal to the S_pore_ value. Therefore the surface of the pores was calculated from the surface of the material. Based on isotherm and pore parameters, it could be concluded that ZVI is a nonporous material.

Another essential part of the work was the material composition analysis using an XRF analyzer (Figure 2). The results obtained showed the lack of additional ingredients in the composition of the ZVI and thus its excellent purity.

The results confirmed that the selected ZVI could be successfully used as a heterogeneous catalyst, a potential source of iron for the UV/ZVI/H_2_O_2_ process.

### 3.2. UV/ZVI/H_2_O_2_ Process

The initial concentration of dyes was equivalent to that observed in real industrial wastewater, 50 mg L^−1^. The organic compound content of the solution was verified using TOC. The initial TOC for AM ACID was 14.57 mg L^−1^ and for AM E123 it was 20.58 mg L^−1^. This led us to the conclusion that the additives in the composition of AM ACID were inorganic. The initial light absorbance was 3.097 for AM ACID and 3.118 for AM E123.

Preliminary tests were performed to preselect the range of reagent doses for use in the main experiment. The catalyst dose was selected after 15 min of sample irradiation with H_2_O_2_. The important element of the optimization was a dose of the catalyst—ZVI. An essential part of the research involved selecting the appropriate amount of solid catalyst to ensure that its oxidation provided a sufficient dose of Fe^2+^ ions.

The preliminary tests were conducted by irradiating the sample for 15 min with 400 mg L^−1^ H_2_O_2_. The catalyst dose at which the TOC result decreased the most was selected for use in the main experiment. Therefore, 500 mg of ZVI to decompose AM E123 and 100 mg to remove AM ACID were selected. In both cases, using a greater amount of catalyst did not result in increased dye removal efficiency, and in fact, an excess of ZVI in the solution was observed to have an inhibitory effect (12), (13).
Fe^2+^ + HO^●^ → Fe^3+^ + HO^−^(12)
Fe^2+^ + HOO^●^ → Fe^3+^ + H_2_O_2_(13)

Various doses (200, 400, and 800 mg of L^−1^) H_2_O_2_ were tested (Figure 3). The use of 800 mg L^−1^ of H_2_O_2_ inhibited the process through undesirable radical reactions. However, regardless of the H_2_O_2_ dose, complete decolorization of the solutions occurred during the process.

The next step of the experiment based on verifying the correctness of the assumption previously made regarding the optimality of pH 3. After conducting the processes at pH 3, 4, 5, and 6 decreases in process efficiency were observed from 82.5% to 15.78% and from 80.85% to 11.37% for AM ACID and AM E123, respectively. As the pH of the solution increased, the potential of hydroxyl radicals decreased. Therefore, no radical mechanism occurred in neutral or alkaline environments, and iron (IV) species appeared (reactions (14) and (15)).
Fe^2+^ + H_2_O_2_ → FeO^2+^ + H_2_O(14)
FeO^2+^ + H_2_O_2_ → Fe^2+^ + H_2_O + O_2_(15)

In near-neutral and alkaline environments, the coagulation of Fe^3+^ ions began, limiting the catalytic properties of Fe^2+^ ions. In addition, the dissociation of H_2_O_2_ to oxygen and water began. Furthermore, as the pH increased, limited or complete lack of decolorization was observed.

The experiments performed confirmed the earlier predictions and the proper selection of doses.

### 3.3. Reactive Oxygen Species

AOPs are excellent methods for the decomposition of highly persistent organic compounds. The superior results for dye decomposition are produced by the activity of reactive oxygen species (ROS), primarily hydroxyl radicals [30]. However, radicals of lower potential, which can still oxidize contaminants, are formed during the process and its modification. In this work, singlet oxygen (^1^O_2_) was revealed—a less reactive form of ROS. The results are shown in Figure 4. After the ZVI/UV/H_2_O_2_ process was conducted under optimal conditions, the detected fluorescence intensity of the samples was more than 2000% compared to that of the samples before the process. Thus, the high activity of ROS, including singlet oxygen, in samples during the photo-Fenton process, was confirmed.

### 3.4. Process Influence on Catalyst Morphology and Structure

Scientists used novel and active materials with environmentally friendly properties to obtain excellent results from the process based on heterocatalysts. ZVI was chosen from the large family of catalysts because of its high efficiency and potential for reuse. The UV/ZVI/H_2_O_2_ process was a highly efficient method of organic dye decomposition. However, before using the catalysts again, changes in the morphology, structure, and stability of the ZVI were examined. The influence of 60 min of the UV/ZVI/H_2_O_2_ process at various pH values on the properties of the catalyst was also investigated.

Firstly, the surface of the catalyst was analyzed using Fourier transform infrared spectroscopy (FTIR), and its properties after the process were compared with AM ACID and AM E123. The results are shown in Figure 5a,b. The characteristic rocking vibrations for amaranth dyes were detected in the samples before the process. The same energy vibrations were not detected for materials at various pH values after the process. Thus, the pH of the solution and the dye type did not affect the surface properties of ZVI. Furthermore, the lack of additional organic groups suggests the immediate decomposition of the compound without solid adsorption on the surfaces of the catalyst.

In the next step, the influence of various pH values and dyes on the stability of the material was tested based on zeta-potential analysis. The results obtained are shown in Figure 5c,d for AM ACID and AM 123, respectively. The stability was similar for both types of dye before the process. For those processes, a zeta potential of about 10 eV was observed. Interestingly, the lowest stability of the material (1 eV) was noted for the samples after the process, with a pH value of 3, regardless of the type of amaranth.

Furthermore, a zeta potential in the range of 8 to 12 eV was noted in the samples after the decomposition of AM E123 at pH 4 to 6. The best stability was noted for the catalyst after the process with AM ACID at pH 5 and 6, with zeta potential values of 15 and −10 mV detected, respectively. These results suggest that a high pH value and the presence of AM ACID did not significantly affect the ZVI.

Next, the composition, quality, and purity of the catalysts before and after the decomposition of both dyes during the treatment processes were studied. This involved the analysis of the presence of iron in the samples. The results of the analyses are shown in Figure 6. The iron was detected in all the samples regardless of the dye. Therefore, the ZVI could be reused and utilized in the following processes to decompose organic compounds. The material is characterized by greater stability in conditions closer to neutral. This is, among other things, related to the solubility of individual forms of iron. However, in an acidic environment, iron will dissolve, Fe (II) ions will enter the solution, Fe (III) forms may and will appear, resulting from the Fenton reaction. It is believed that the optimal pH for the Fenton process is 3.0. The research aimed at decomposing the dye, which was considered more important than the stability of the material used. The process of dye decomposition itself results from the occurrence of radical reactions that are part of the Fenton process. A side effect of using pH 3.0 during the process is the partial destruction of the catalyst and the need to separate Fe (II) ions from the treated wastewater—it was done by causing the final coagulation, raising the pH to 9.0 after a certain time of the process, which also allows the removal of unreacted hydrogen peroxide.

Finally, the influence of the process on the morphology of the catalyst, particularly its size and shape, was investigated. Thus, the mean diameter, circularity, and agglomeration of materials after the decomposition of dyes were analyzed and are shown in Figure 7 and Figure 8.

The highest hydrodynamic diameter of the grains was detected for samples at pH 4 (1000 to 3500 nm) after the process with AM ACID. Similar results were observed for samples before the process and at pH 3 and pH 5 (from 750 to 2500 nm), and the lowest value was observed at pH 6 (from 500 to 1500 nm) for AM ACID. For samples after the process with AM E123, the smallest hydrodynamic diameter of the grains before the process (from 500 to 2000 nm) were detected, and these were slightly bigger after the process at pH 3 and 4 (from 1000 to 2500 nm.) The highest diameter was noted for samples after the process with AM E123 as well as at pH 4 and 6 (from 1500 to 4000 and 2000 to 5000 nm, respectively). In addition, the circularity results for all the samples were identical and in the range from 0.3 to 0.4.

Additionally, for all samples, a similar equivalent circular area diameter (in the range from 5 to 30 μm) was detected. However, for both types of samples, different peak intensities were noted: a lower intensity for samples before the process compared with after the process (0.4 vs. 1.4%). The results obtained suggested a slight increase in diameter caused by the mechanical aspects of the process, such as the stirring. In addition, grain agglomeration increased after the decomposition of both types of dye.

### 3.5. ZVI Diffusion Test Results

The next part of this work involved the diffusion test. These studies were conducted to assess the stability of ZVI when used as a catalyst. Therefore, pristine ZVI (before the process) and ZVI after treatment of both dyes at pH 3 were investigated. The main principle of the test involved the measurement of diffusion zones. In this study, such zones resemble ginger-colored circles, which result from the oxidation of the samples during incubation, and subsequent diffusion of iron ions into an agar medium. If the process did not influence the stability of the material, the size of the diffusion zones around the utilized species of the microorganism should be similar. The results of the diffusion tests are presented in Figure 9.

According to the results obtained, the catalyst was highly stable and not affected by the treatment process. The most significant diffusion zones were in the samples with *E. coli* bacteria, and the smallest ones were in the samples incubated in the presence *of C. albicans*. The diffusion zones were characterized by their similar size around all the investigated microorganisms, which suggested the identical potential of pristine material and that following treatment in terms of its effect on the studied bacteria and yeast. Therefore, it can be concluded that the treatment process did not affect the stability of the material used as the catalyst.

### 3.6. Reusability

A very important part of the research was to verify the reusability of the catalyst. As with process optimization, the possibility of catalyst reuse not only improves the process but also reduces its cost. After each run, the ZVI was magnetically separated and placed in a fresh portion of the dye solution. The ZVI was subjected to a triple dye treatment cycle with little fluctuation in process efficiency. In the case of AM ACID, the efficiency of all three cycles oscillated around an 81% decrease in the TOC amount. The TOC in AM E123 degradation, on the other hand, decreased with each cycle from nearly 82.5 to 76% (Figure 10).

Nevertheless, the catalyst stability results were positive. The slight differences in efficiency after successive processes testify to the effective oxidation capacity of the molecules. In addition, decolorization was maintained at more than 99% in all processes.

## 4. Discussion

### 4.1. ZVI Properties

Because of its low S_BET_ value, ZVI does not demonstrate the properties of a good sorbent. The pH changes during AOP have no influence on the sorption properties of the catalyst, which was confirmed by the absence of organic compounds on the catalyst surface. The morphological changes of the catalyst during the process indicated that the release of iron ions into the solution occurred in parallel with catalyst grain agglomeration during the stirring process. Despite the change in the degree of dispersion, the dye decomposition process was very effective.

### 4.2. UV/ZVI/H_2_O_2_ Process

UV/H_2_O_2_ is a catalyst with UV irradiation, and tests of UV irradiation alone were carried out to assess the effect of each factor on process efficiency. These processes were conducted for both dyes. The highest process efficiencies were observed when all the factors were used together in a modified Fenton process. The decreased efficiency noted after the individual tests were related to the occurrence of minor processes during the tests, i.e., photolysis, adsorption, and oxidation. In addition, the absorbance of the samples oscillated around three after tests using a catalyst with UV irradiation and using UV irradiation alone for both dyes. The greatest decolorization of the solutions was achieved using the UV/H_2_O_2_ process, but even in that case, the solutions were not fully decolorized. This suggests that the synergistic action of all factors, in addition to the minor processes mentioned above, supports the generation of radicals that are capable of oxidizing pollutants.

In order to maintain the high effectiveness of processes and lower the costs related to the purchase of the reactants, the optimal doses of these reactants were determined. A series of tests were performed to determine the most effective doses of each reagent. Preliminary tests were conducted at pH 3, which is widely accepted to be optimal for ZVI use. The doses of the reagents were carefully selected based on the results of the TOC.

To investigate the mechanism of the process, the reaction kinetics were described. The following equations were used ((16)–(19)):TOC = TOC_0_ + e^−kt^(16)
TOC = (kt + 1/TOC_0_)^−1^(17)
TOC = (TOC_0_ − b) + e^−kt^ + b(18)
TOC = (kt + (TOC_0_ − b)^−1^)^−1^ + b(19)
where TOC_0_ is the initial TOC value of the dye solutions, t is the time of the process, k is the photo-decolorization reaction rate constant, and b is the process-correcting parameter for nonoxidizable substances. For each experiment related to dye removal using the UV/ZVI/H_2_O_2_ process, the compliance of the process with the model of first-order kinetics (16), modified first-order kinetics (18), second-order kinetics (17), and modified second-order kinetics (19) was calculated. Therefore, a figure for each experiment was obtained, which showed the actual course of the process vs. the course for each of the four models. The reaction that reflected the best results for dye degradation was the reaction (19) called the modified pseudo-second-order reaction. Because of the large number of figures, only one example is shown (Figure 11).

The method’s capacity for dye treatment with regard to the TOC value was analyzed based on the results. The fastest decrease in TOC value occurred in the first 15 min. After that, the process slowed down, and the reaction curve flattened out. However, the optimal process time was 60 min.

The ANOVA method was used to study the magnitude of variability in the average TOC concentrations. TOC value depended more on the processing time than the dose of H_2_O_2_. This is because of the time required for the regeneration of Fe^2+^ ions and for H_2_O_2_ to react with the available Fe^2+^ ions. However, time had a less significant effect on process efficiency than pH (Figure 12 and Figure 13).

Based on the results obtained, it could be concluded that UV/ZVI/H_2_O_2_ may be effective when used to decompose amaranth dyes. The efficiency of amaranth removal was comparable to results obtained in research that used UV/ZVI/H_2_O_2_ to remove other dyes, i.e., rhodamine B [20], Congo red [22], and methylene blue [23]. In a study on rhodamine B removal, Liang et al. achieved almost complete dye degradation using doses of reagents (Fe^0^ = 9 mM, H_2_O_2_ = 8 mM) similar to those described in this paper. The researchers also noted that the use of higher doses of H_2_O_2_ led to process inhibition. In addition, it was concluded that complete decomposition of the dye does not occur. The decomposition rate of dye removal was higher than the decrease in the TOC value [20].

In contrast, Lui et al. obtained a lower removal efficiency with higher dye concentrations compared with the results obtained in this study. Furthermore, the processing time was longer, which is disadvantageous from an application viewpoint. However, almost 99% color removal was observed after the first 10 min of the process at a catalyst dosage of 100 mg L^−1^. The color decay was due to the breakage of the -N=N- bond. Because amaranth dyes belong to the same group as Congo red, these bonds are also present in their structure. Thus, the color decay in our research also occurred because of the aforementioned bond breakage [22].

Another study focused on the decomposition of methylene blue (MB), a thiazine dye, using dissolved copper in a ZVI/H_2_O_2_ process with which Yang et al. obtained satisfactory results. In contrast, the same process without the addition of copper ions was shown to be ineffective. This does not contradict our results, as Yang conducted research at a pH of 5.5. The dye decomposition using UV/ZVI/H_2_O_2_ occurs at a much lower pH, which was also reflected in our research [23].

## 5. Conclusions

In this work, the feasibility of using a commercial micron-sized ZVI, Hepure Ferox Flow, to treat synthetic dyes used in the cosmetics industry was tested. The excellent properties of ZVI, such as the large specific surface absorbance and direct bandgap, indicated that the material was suitable for photo-driven catalytic treatment processes.

The UV/ZVI/H_2_O_2_ process was shown to be a highly efficient method of decomposing two organic dyes. The optimal process conditions determined for the removal of AM E123 and AM ACID, both with a concentration of 50 mg L^−1^, were 500 and 100 mg L^−1^ of ZVI, respectively, an H_2_O_2_ dose of 400 mg L^−1^, a pH of 3, and a processing time of 60 min under UV irradiation. The process efficiency under these optimal conditions was 85.5 and 80.85% for AM E123 and AM ACID, respectively. Complete decolorization was observed in all samples. The greatest TOC decrease in the AM ACID decomposition process was found at a lower ZVI dose compared with AM E123, because of the inorganic additives in AM ACID. ZVI is an effective catalyst for the decomposition of both dyes using the UV/ZVI/H_2_O_2_ process.

The decomposition of both dyes occurred according to the modified pseudo-second-order reaction. The fastest TOC decrease was obtained in the first 15 min, and further dye decomposition slowed down. There was a statistically significant correlation between the TOC decrease, pH, and process time.

The high stability of the catalyst was observed, and it was not affected by the treatment process even after the third cycle of the process, as confirmed by the results of the catalyst surface analysis and iron diffusion test. Additionally, only slight differences were observed in process efficiency after each cycle.

The need for only a small amount of catalyst to decompose AM E123 and AM ACID and the additional ability to reuse the catalyst without the need for prior preparation may enable cost savings in catalyst purchase.

This research is the first stage of work to explore the possibility of using the UV/ZVI/H_2_O_2_ method for dye decomposition. In this stage, process parameters, including pH value and reagent doses for aqueous solutions of dyes, were determined. In the second stage, which is a continuation of the current work, the treatment of real wastewater containing dyes is planned.

## Figures and Tables

**Figure 1 materials-16-01523-f001:**
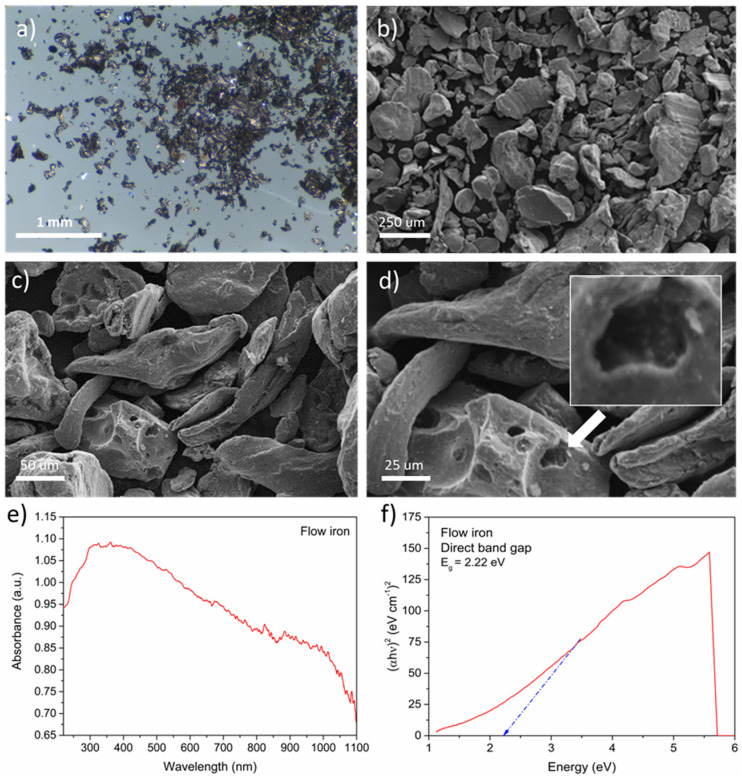
ZVI characterization: (**a**) digital images, (**b**–**d**) SEM images, (**e**) absorbance, and (**f**) direct band gap.

**Figure 2 materials-16-01523-f002:**
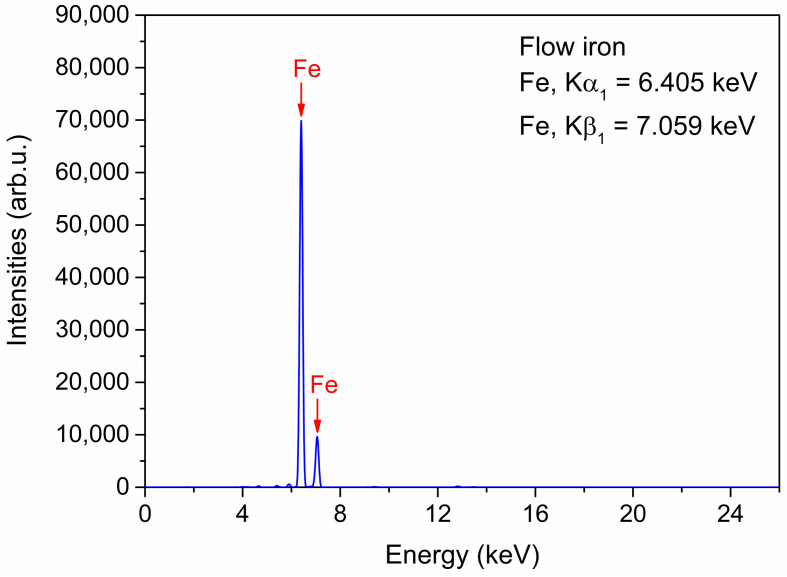
The XRF analysis of the purity of the ZVI.

**Figure 3 materials-16-01523-f003:**
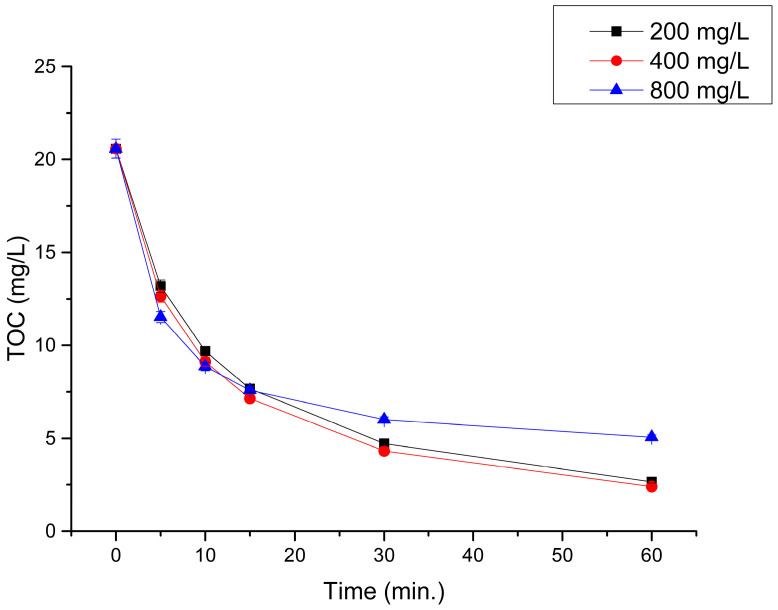
AM E 123 decomposition through the UV/ZVI/H_2_O_2_ process, for selected H_2_O_2_ doses and 500 mg L^−1^ ZVI, pH = 3.0.

**Figure 4 materials-16-01523-f004:**
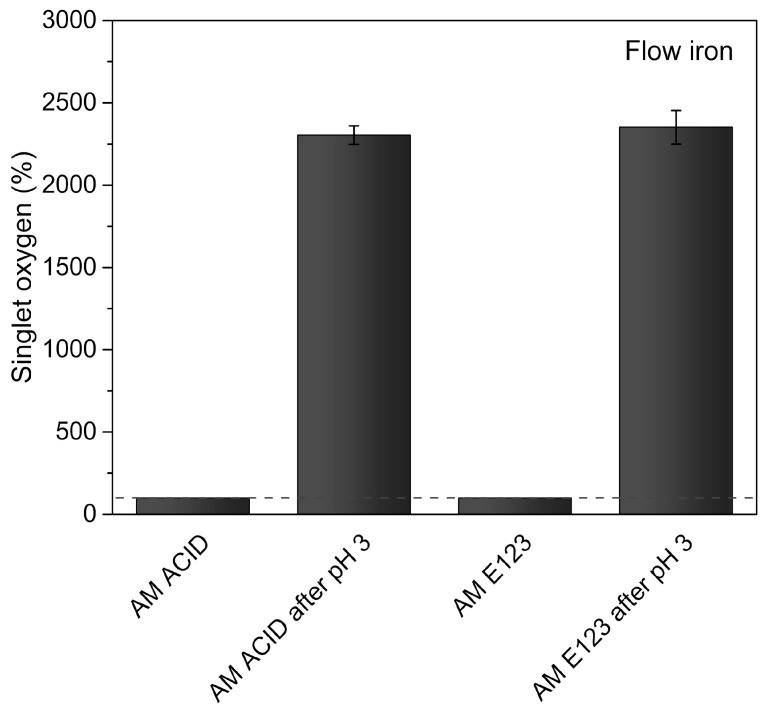
Singlet oxygen content during the UV/ZVI/H_2_O_2_ process.

**Figure 5 materials-16-01523-f005:**
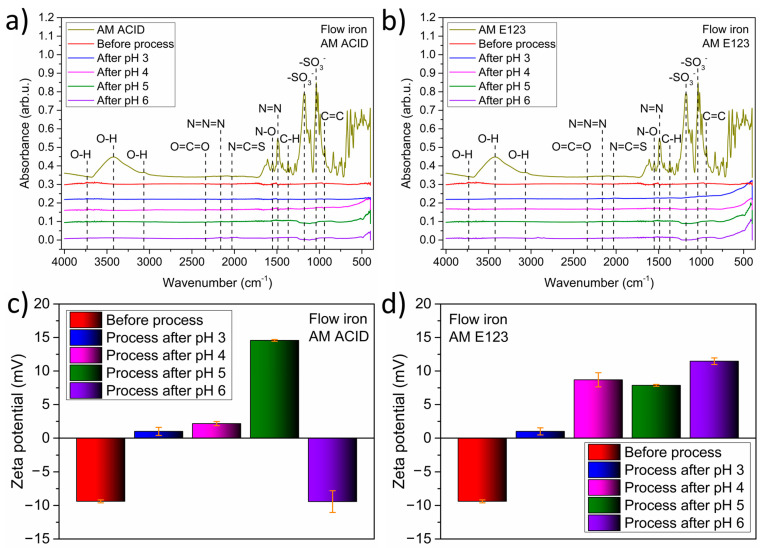
Catalyst surface analysis with (**a**) AM ACID and (**b**) AM E123, and stability with (**c**) AM ACID and (**d**) AM E123 after 60 min of the UV/ZVI/H_2_O_2_ process.

**Figure 6 materials-16-01523-f006:**
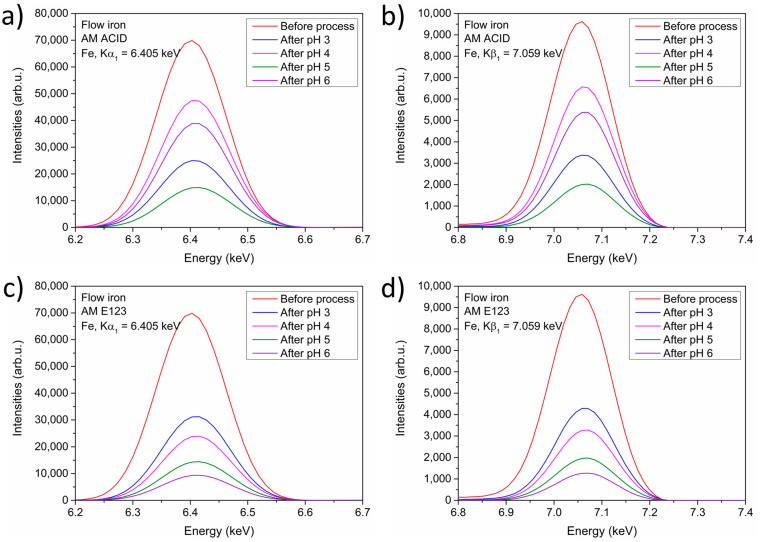
XRF analysis of the presence of iron in Kα_1_ = 6.405 keV after the process with (**a**) AM ACID and (**c**) AM E123. XRF analysis of the presence of iron in Kβ_1_ = 7.068 keV after the process with (**b**) AM ACID and (**d**) AM E123.

**Figure 7 materials-16-01523-f007:**
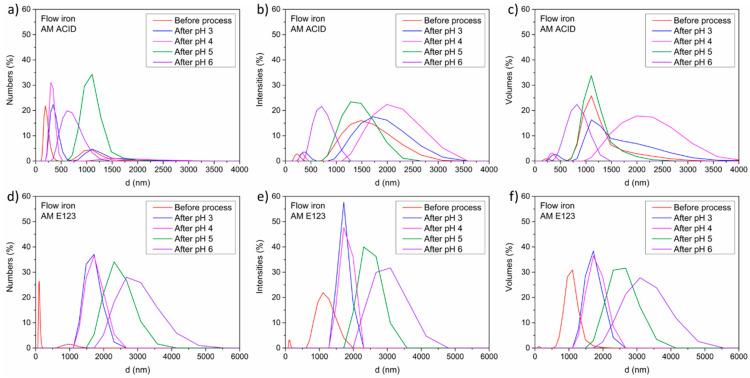
Size characterization results for the catalysts after the process with numbers of the particles (**a**), intensities (**b**), and volumes (**c**) of AM ACID and numbers of the particles (**d**), intensities (**e**), and volumes (**f**) of AM E123.

**Figure 8 materials-16-01523-f008:**
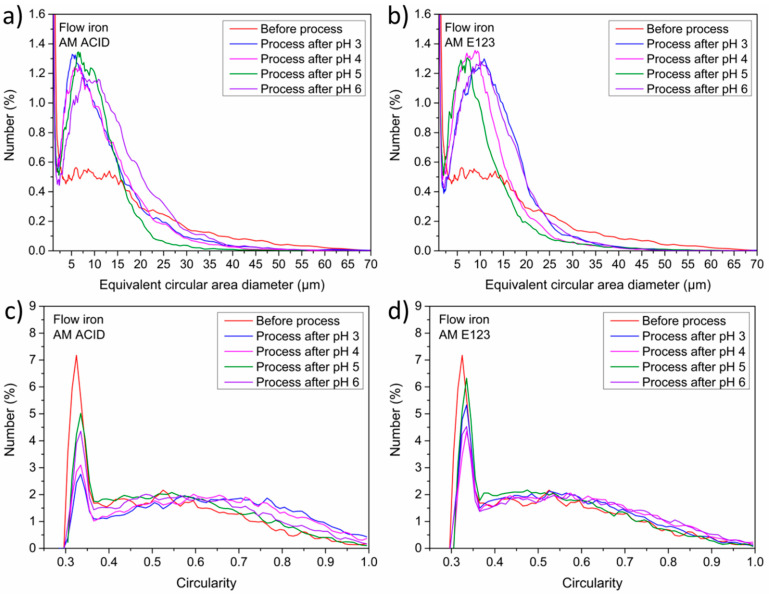
Comparison of shape parameters such as minimum equivalent circular area diameter for samples after the process with (**a**) AM ACID and (**b**) AM E123. Circularity of the material’s grain after methods with (**c**) AM ACID and (**d**) AM E123.

**Figure 9 materials-16-01523-f009:**
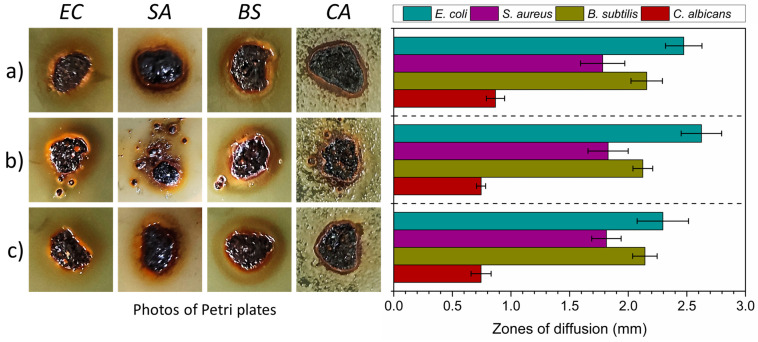
The results of diffusion studies performed for (**a**) reference ZVI, (**b**) ZVI after treatment of AM ACID at pH 3, and (**c**) ZVI after treatment of AM E123 at pH 3. The tests were performed with *E. coli* (EC), *S. aureus* (SA), *B. subtilis* (BS) bacteria, and yeast *C. albicans* (CA). The results are presented as photographs of Petri plates showing zones of diffusion (left side of the figure), and measured zones of diffusion (right side of the figure).

**Figure 10 materials-16-01523-f010:**
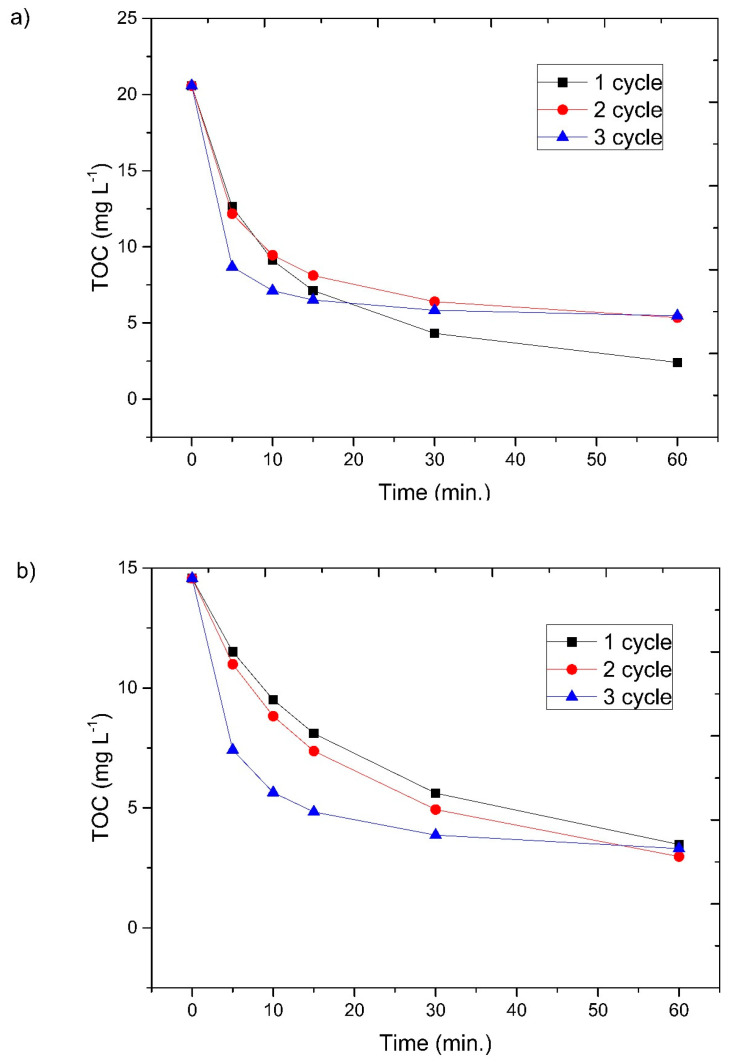
Reusability for three cycles with optimal parameters of 400 mg L^−1^ H_2_O_2_, pH 3, and UV irradiation: (**a**) with 500 mg ZVI and 50 mg L^−1^ AM E123 and (**b**) with 100 mg ZVI and 50 mg L^−1^ AM ACID.

**Figure 11 materials-16-01523-f011:**
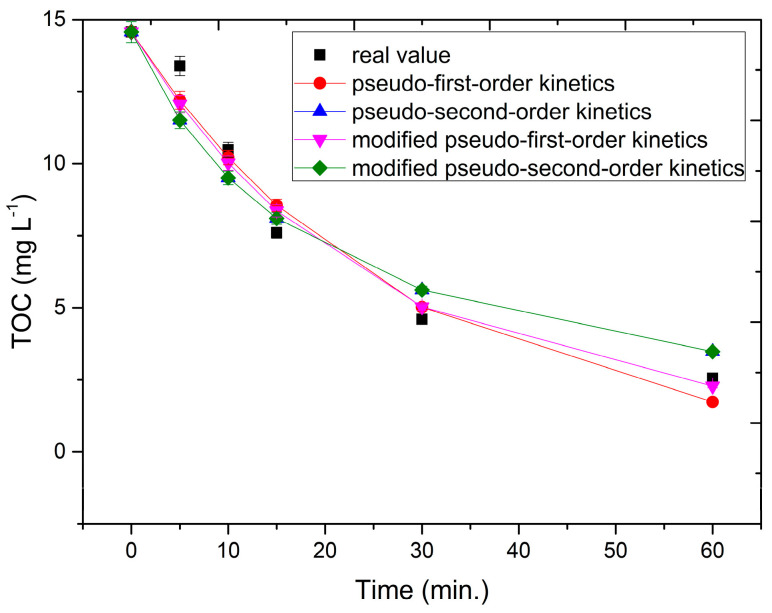
Kinetic model fitting for AM ACID, H_2_O_2_ 400 mg L^−1^, and ZVI 100 mg L^−1^.

**Figure 12 materials-16-01523-f012:**
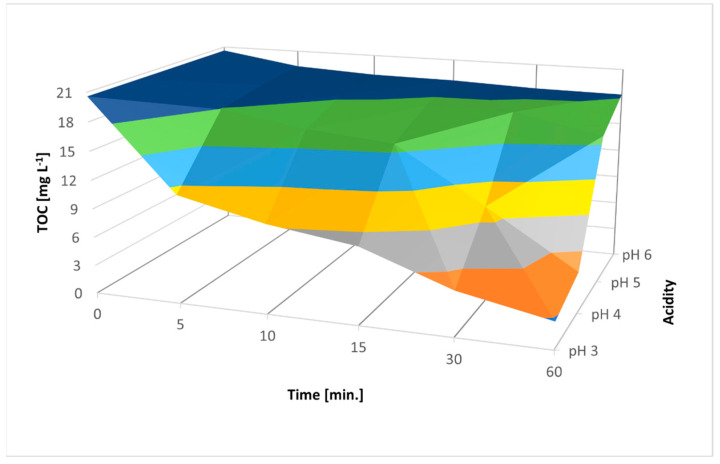
ANOVA results for different pH values with optimal parameters of 400 mg L^−1^ H_2_O_2_ and UV irradiation with 500 mg ZVI and 50 mg L^−1^ AM E123.

**Figure 13 materials-16-01523-f013:**
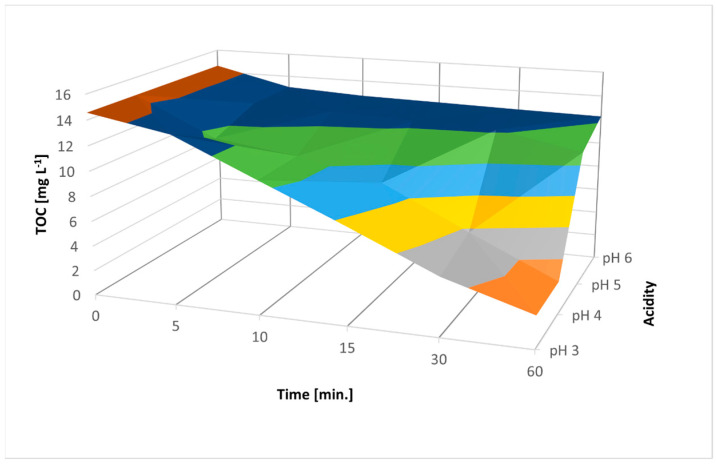
ANOVA results for different pH values with optimal parameters of 400 mg L^−1^ H_2_O_2_ and UV irradiation with 100 mg ZVI and 50 mg L^−1^ AM ACID.

## Data Availability

Data and materials that support the findings of this study are available from the corresponding author upon reasonable request.

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
