# Peer review of "Application of Micron-Sized Zero-Valent Iron (ZVI) for Decomposition of Industrial Amaranth Dyes"

_materials, 2023, doi:10.3390/ma16041523_

Round 1

Reviewer 1 Report

See the attachment.

Author Response

Dear Reviewer,

We would like to thank the Reviewer for the critical reading, as well as for helpful, relevant, and constructive remarks. The manuscript was corrected and rewritten in accordance with the suggestions, which improved the quality of our paper. All changes are indicated in red. We acknowledge that these modifications definitely improve the quality of our manuscript. We hope that the changes and explanations are acceptable and satisfactory with the expectation of the Reviewer.

You can find below the details of the modifications and explanations. Thank you very much for revising our manuscript again!

General Comment: This manuscript aims to examine whether a micro zero valent iron (mZVI), ferox flow, can be used for the effective removal of dyes used in the industry. It also studies the microstructures of the ZVI and attempts to explain the mechanism of how the ZVI works to decompose the dyes when UV radiation hydrogen peroxide are present. In addition, the stability of such ZVI was also investigated. The results do provide additional insight into finding efficient and cost-effective methods for the treatment of organic pollutants in water system. However, there are a couple of major issues which need clarifications/revisions prior to the manuscript being considered for publication.

Answer to general comment 1: Thank you for the positive reception of our manuscript. We have made every effort to improve the manuscript and to clarify all ambiguities.

Comment 1: Supplementary document is NOT included in the submission. The document named as “materials-2129550-supplementary” is the manuscript in Doc format, no the

supplementary file.

Answer to comment 1: Thank you for paying attention. We are very ashamed of having made this mistake. However, due to other comments from Reviewers, it was decided to move these materials to the main article. There were, among others, graphs regarding the course of the UV/ZVI/H2O2 process and kinetics. There are many graphics that were added to main body of our article. The total number of figures in the article after the changes is 14. Therefore, in order not to unnecessarily extend the response for the Reviewer 1, we do not include them here. We kindly ask the Reviewer 1 to check directly in the corrected version of the article.

Comment 2: In the sections of “3.2. UV/ZVI/H2O2 process” and “3.3. Kinetics calculations”, no single graph or table is given. Therefore, it is impossible for readers to perceive the claims

made by the authors. Supporting data are needed.

Answer to comment 2: We agree with the Reviewer that it is worth enriching these chapters with graphics, as we mentioned in answer to comment 1.

In the case of section 3.2, experiments were performed on the effect of different doses of hydrogen peroxide on the effectiveness of TOC removal. The results can be presented on several figures, but due to the fact that the scope of our research is quite large, which results in many different results, we decided to supplement the article with one example figure illustrating selected results of TOC removal in the UV/ZVI/H2O2 process. The figure shows repeated trends from which optimal doses were determined.

A similar situation appears for section 3.2 for each experiment related to dye removal using the UV/ZVI/H2O2 process, the compliance of the process with the model of first order kinetics, modified first order kinetics, second order kinetics and modified second order kinetics was checked. Therefore, we have prepared a graph for each experiment, which shows the actual course of the process vs. the course for each of the 4 models. It was determined that the model of modified second-order kinetics best reflects the actual course of the process. Due to the large number of graphs, each of which would present exactly the same information, we decided to include only one sample graph in the article.

Comment 3: A few more minor issues are also needed to be addressed.

  1. The format of Abstract is a common one for this journal. It is better to write it in an integrated format, not divided into different sections.
  2. In “Introduction”, it is recommended that the authors stress the novelty of their research, which is not clearly stated.
  3. In “3.1. ZVI characterization”, please show the pore size distribution so that the claim about the average pore size can be supported.

Answer to comment 3a: Thank you, for pointing this out. We have made every effort to ensure that the abstract reflects the contents of the manuscript as closely as possible. However, we made some modifications to the abstract to make it even more precise. Words: background, methods, results and conclusions were removed from abstract, to obtain an integrated format, not divided into different sections.

Answer to comment 3b: With reference to the comments of all the Reviewers, we rewrite the introduction paragraph. Cited literature has been clarified. The scientific novelty and the aim of this article was clearly marked and at the same time the difference of this article from other previously published articles was indicated.

Answer to comment 3c: In paragraph 3.1 the pore size distribution and other results characterizing the surface properties of the ZVI have been added for confirmation statements contained in the article.

We have added following paragraph:

The physical properties of the ZVI were obtained using the physical nitrogen sorption method using the BJH (Barrett–Joyner–Halenda) method and the single point BET model. The results revealed that the surface area SBET had a value of 1.711 m2 g-1. The average pore surface Spore was 1.507 m2 g-1 pore volume, Vpore 0.003 cm3 g-1, and the pore diameter Dpore was 20.201 nm. The SBET value was practically equal to the Spore value, therefore the surface of the pores was calculated from the surface of the material. Based on isotherm and pore parameters, it could be concluded that ZVI is a nonporous material.

In order to ensure the appropriate linguistic quality of the article, it has been submitted to the MDPI English Editing Service for linguistic corrections.

Once again, thank you very much for your comments and suggestions.

Reviewer 2 Report

If this ZVI material can be used for the decomposing of other dyes, such as rhodamine B, and Congo red?

If the decomposition can be affected by the temperature? During the UV/ZVI/H2O2 process, what's the temperature? if different temperatures were considered?

Author Response

Dear Reviewer,

We would like to thank the Reviewer for the critical reading, as well as for helpful, relevant and constructive remarks. The manuscript was corrected and rewritten in accordance with the suggestions, which improved the quality of our paper. All changes are indicated in red. We acknowledge that these modifications definitely improve the quality of our manuscript. We hope that the changes and explanations are acceptable and satisfactory with the expectation of the Reviewer.

You can find below the details of the modifications and explanations. Thank you very much for revising our manuscript again!

Comment 1: If this ZVI material can be used for the decomposing of other dyes, such as rhodamine B, and Congo red?

Answer to comment 1:

Most of the dyes used practically in industry, including Rhodamine b or Congo red, examples mentioned by Reviewer 2, are organic compounds. Advanced oxidation processes are designed to produce radicals, non-selectively oxidizing various organic compounds. On this basis, it can be assumed that the ZVI we used can be used to remove the aforementioned dyes. To confirm this assumption, we performed a literature review on the possibility of using ZVI to remove various dyes. We added following paragraph:

Based on the results obtained, it could be concluded that UV/ZVI/H2O2 may be effective when used to decompose of amaranth dyes. The efficiency of amaranth removal was comparable to results obtained in research that used UV/ZVI/H2O2 to remove other dyes, i.e., rhodamine B [20], Congo red [22], and methylene blue [23]. In a study on rhodamine B removal, Liang et al. achieved almost complete dye degradation using doses of reagents (Fe0 = 9mM, H2O2 = 8 mM) similar to those described in this paper. The researchers also noted that the use of higher doses of H2O2 led to process inhibition. In addition, it was concluded that complete decomposition of the dye does not occur. The decomposition rate of dye removal was higher than the decrease in the TOC value. [20]

In contrast, Lui et al. obtained a lower removal efficiency with higher dye concentrations compared with the results obtained in this study. Furthermore, the processing time was longer, which is disadvantageous from an application viewpoint. However, almost 99 % color removal was observed after the first 10 minutes of the process at a catalyst dosage of 100 mg L-1. The color decay was due to the breakage of the -N=N- bond. Because amaranth dyes belong to the same group as Congo red, these bonds are also present in their structure. Thus, the color decay in our research also occurred because of the aforementioned bond breakage. [22]

Another study focused on the decomposition of methylene blue (MB), a thiazine dye, using dissolved copper in a ZVI/H2O2 process with which Yang et al. obtained satisfactory results. In contrast, the same process without the addition of copper ions was shown to be ineffective. This does not contradict our results, as Yang conducted research at a pH of 5.5. The decomposition of dye using UV/ZVI/H2O2 occurs at a much lower pH, which was also reflected in our research. [23]

Comment 2: If the decomposition can be affected by the temperature? During the UV/ZVI/H2O2 process, what's the temperature? if different temperatures were considered?

Answer to comment 2:

UV/ZVI/H2O2 processes could be affected by temperature. As the temperature increases, the efficiency of the contaminant degradation may increase. However, too low and too high temperatures both have an inhibitory effect on the processes. In our study, the processes were carried out only at room temperature (about 20 °C) only. We conducted our research in cooperation with an industrial partner, in terms of the possibility of a potential application. The choice of room temperature was related to the method of conducting the process in the plant. Reducing/increasing the wastewater temperature would require heating/cooling, which would increase costs. However, we agree with the Reviewer 2 that it is interesting from a scientific point of view to accurately determine the effect of temperature on the effectiveness of the process. We will take this into account in the next part of our research, which will deal with the removal of dyes from real industrial wastewater.

In order to ensure the appropriate linguistic quality of the article, it has been submitted to the MDPI English Editing Service for linguistic corrections.

Once again, thank you very much for your comments and suggestions.

Author Response

Dear Reviewer,

We would like to thank the Reviewer for the critical reading, as well as for helpful, relevant and constructive remarks. The manuscript was corrected and rewritten in accordance with the suggestions, which improved the quality of our paper. All changes are indicated in red. We acknowledge that these modifications definitely improve the quality of our manuscript. We hope that the changes and explanations are acceptable and satisfactory with the expectation of the Reviewer.

You can find below the details of the modifications and explanations. Thank you very much for revising our manuscript again!

Comment 1: The title seems ok.

Answer to comment 1: Thank you, we have made every effort to ensure that the title of the article reflects the content of the manuscript as closely as possible.

Comment 2:  The abstract seems to be good. I really appreciate to author to well organize and

explained precisely the abstract.

Answer to comment 2: Thank you, we have made every effort to ensure that the abstract reflects the contents of the manuscript as closely as possible. However, at the request of Reviewer 1, we made some modifications to the abstract to make it even more precise. Words background, methods, results and conclusions were removed from abstract, to obtain an integrated format, not divided into different sections. We hope that the modifications we have made will be appreciated by Reviewer 3.

Comment 3:  Remove this keyword amaranth E123.

Answer to comment 3: Keyword amaranth E123 was removed.

Comment 4: Also changed this keyword from advanced oxidation processes to oxidation processes.

Answer to comment 4: Keyword advanced oxidation processes was changed to oxidation processes. Word “advanced” was removed.

Comment 5: Research gap should be delivered on more clear way with directed necessity for the future research work.

Answer to comment 5: We are conducting research in collaboration with an industrial partner using dyes whose removal was studied in this manuscript. There are problems related to the treatment of wastewater containing dyes (too high concentrations in wastewater). Therefore, it is necessary to remove dyes from wastewater with more efficient method. The dyes doses selected in current research coincide with the doses observed in wastewater. The current research is the first stage of work in which we determine the possibility of using the UV/ZVI/H2O2 method and also pre-select doses for the process. In the second stage, which will be a continuation of the current work, treatment of real wastewater containing dyes is planned.

Comment 6: Introduction section must be written on more quality way, i.e., more up-to-date

references addressed.

Answer to comment 6: With reference to the comments of all the Reviewer 3, we rewrite the introduction section. Cited literature has been clarified.

Comment 7: The novelty of the work must be clearly addressed and discussed, compare previous research with existing research findings and highlight novelty.

Answer to comment 7: The novelty of the research was stated at the end of introduction section. We also point out difference between our research and previously published ones.

Comment 8: What is the main challenge?

Answer to comment 8: The presence of dyes in industrial wastewater and water is a serious environmental and social problem. Therefore, they must necessarily be removed from the aqueous phase. These compounds are resistant to removal by conventional methods such as sorption or coagulation. Therefore, AOP seems to be an interesting alternative.

Comment 9: Please check the abbreviations of words throughout the article. All should be

consistent.

Answer to comment 9: All abbreviations were checked, explained, corrected and unified.

Comment 10: What is problem statement? State clearly.

Answer to comment 10: Thank you for pointing this out. Most research on the application of ZVI focuses on the fabrication of the nanomaterial. Such a structure will have excellent catalytic properties but will be very expensive to produce, which makes it impossible to use it in industrial processes. On the other hand, waste iron usually has a very large grain size and therefore a small active surface. The main challenge of our research was to determine whether the cheap commercial product Hepure Ferrox Flow available on the market (this one is dedicated to the treatment of groundwater in which amount of pollutants is considered to be small) with a grain size larger than nano-size are effective enough to be considered as a potential catalyst in UV/ZVI/H2O2 process.

Comment 11: There are several irrelevant references please take a strong revision in this section.

Answer to comment 11: Irrelevant references were removed. In their place, important references have been supplemented. References have been supplemented with items that are current and relevant to the subject of the article.

Comment 12: Replace reference 1 with this reference, the stated information is well explained in this article. - Role of nanomaterials in the treatment of wastewater: A review.

Answer to comment 12: Publication: Yaqoob, A.A., Parveen, T., Umar, K., Ibrahim, M.N.M. Role of nanomaterials in the treatment of wastewater: A review. Water 2020, 12(2), 495. https://doi.org/10.3390/w12020495 was added to cited literature as cited item 1. Publication 1 in the first version of the article has been removed from the list of cited literature. 

Comment 13: Second remove reference 2- 6 and at the end of sentence simply cite this single

reference- Recent advances in new generation dye removal technologies: novel search for approaches to reprocess wastewater.

Answer to comment 13: References 2-6 have been removed. New item (Ahmad, A.; Mohd-Setapar, S.H.; Chuong, C.S.; Khatoon, A.; Wani, W.A.; Kumard, R.; Rafatullah, M. Recent advances in new generation dye removal technologies: novel search for approaches to reprocess wastewater. RSC Adv. 2015, 5, 30801-30818., https://doi.org/10.1039/C4RA16959J) was added. Citations were corrected.

Comment 14: Again remove 7 to 10 reference number and cite single here from latest literature.

Answer to comment 14: References 7-10 have been removed. New item (Mojiri, A.; Bashir, M.J.K. Wastewater Treatment: Current and Future Techniques. Water 2022, 14(3), 448. https://doi.org/10.3390/w14030448) was added. Citations were corrected.

Comment 15: The main objective of the work must be written on the more clear and more concise way at the end of introduction section.

Answer to comment 15: The main objective was presented in more concise way at the end of the introduction. Entire introduction chapter was rearranged.

Comment 16: Add dye specification in methodology part. We rearranged 2.1 paragraph as follows:

In this work, ZVI was used as a catalyst in a modified Fenton process to remove two industrially used dyes. The first dye, Amaranth E123 (AM E123, synonyms: Acid Red 27, Azorubine S, Food Red 9; Food Colours Perczak Sp. J., Piotrków Trybunalski, Poland), is 100% Trisodium 2-hydroxy-1-(4-sulfonato-1-naphthylazo)naphthalene-3,6-disulfonate with empirical formula C20H11N2Na3O10S3. It is commonly used as a food coloring agent and cosmetic dye, as well as in other industries. [24] Exposure to the dye may result in ir-ritation of the eyes, skin, and respiratory system. The second dye, Acid Amaranth (AM ACID; Boruta-Zachem Kolor S.A., Bydgoszcz, Poland), is a mixture of monoazo dyes, identified as Acid Red 27 with different additives which contain fillers and application enhancers. This dye has industrial applications and is mainly used in the textile industry for dyeing protein fibers (such as wool and natural silk), wood, leather, and household chemicals. Exposure may lead to respiratory system and eye irritation, mainly by me-chanical means. Furthermore, temporary corneal staining is possible. Both dyes are uti-lized and provided by a cosmetic factory located in central Poland, which uses them as reagents in the production of cosmetics.

Answer to comment 16: Dye specification, manufacturer and usage was added.

Comment 17: Do not use lumpy reference. It’s very weird. The author should use maximum 2-3 reference at one place. Please revise your paper accordingly since some issue occurs on several spots in the paper.

Answer to comment 17: We checked the manuscript thoroughly, removed redundant and lumped citations, also in reference to earlier comments (12-14). In the corrected version of the manuscript, we do not cite more than two references in one place.

Comment 18: Please provide space between number and units. Please revise your paper accordingly since some issue occurs on several spots in the paper.

Answer to comment 18: Units have been separated from numbers.

Comment 19: Please include all chemical/instrumentation brand name and other important

specification.

Answer to comment 19: We tried to include in all places all chemical/instrumentation brand names and other important information. However, due to confidentiality agreement, we cannot provide details about the cosmetics factory itself.

Comment 20: The unit of temperature is wrongly written. Please revise your paper accordingly since some issue occurs on several spots in the paper.

Answer to comment 20: The method of presenting temperature has been improved and standardized. The temperature is given in degrees Celsius (°C), there is a gap between the value and the unit. This method of recording the temperature was chosen based on a review of the recording of this parameter in other articles published in the Materials journal.

Comment 21: Section 3.1. is very poorly written. Please revise and highlight the main theme.

Answer to comment 21: Section 3.1 was supplemented and revised. Nitrogen adsorption-desorption isotherms and the XRF analysis of purity the ZVI as well as some BET analysis results were added. Figures were corrected. Detailed supplementation is described in answer to comment 22.

Comment 22: Provide the graphs and deep study of BET-specific surface area results.

Answer to comment 22: Suggested data and comments were added.

In addition, the average pore size of the material value was 3.20 nm and the mean pore radius was 2.0 nm, which suggests that the material has excellent adsorption properties (Figure 2).

Figure 2. Nitrogen adsorption–desorption isotherms.

The physical properties of the ZVI were obtained using the physical nitrogen sorption method using the BJH (Barrett–Joyner–Halenda) method and the single point BET model. The results revealed that the surface area SBET had a value of 1.711 m2 g-1. The average pore surface Spore was 1.507 m2 g-1 pore volume, Vpore 0.003 cm3 g-1, and the pore diameter Dpore was 20.201 nm. The SBET value was practically equal to the Spore value, therefore the surface of the pores was calculated from the surface of the material. Based on isotherm and pore parameters, it could be concluded that ZVI is a nonporous material.

Figure 3. The XRF analysis of the purity of the ZVI.

Another essential part of the work was the material composition analysis using an XRF analyzer (Figure 3). The results obtained showed the lack of additional ingredients in the composition of the ZVI and thus its excellent purity.

The results confirmed that the selected ZVI could be successfully used as a heterogeneous catalyst, a potential source of iron for the UV/ZVI/H2O2 process.

Comment 23: Regarding the replications, authors confirmed that replications of experiment were carried out. However, these results are not shown in the manuscript, how many replicated were carried out by experiment? Results seem to be related to a unique experiment. Please, clarify whether the results of this document are from a single experiment or from an average resulting from replications. If replicated were carried out, the use of average data is required as well as the standard deviation in the results and figures shown throughout the manuscript. In case of showing only one replicate explain why only one is shown and include the standard deviations.

Answer to comment 23: Experiments related with dyes decomposition (UV/ZVI/H2O2 process) have been carried out in duplicate. TOC results are given as the average of two measurements. Due to the very simple matrix (water + dye), the dispersion of the results was very small, not exceeding 2.5%. Due to the high convergence of the results and the very small absolute values of the standard deviation, we did not provide these values in the first version of the article. We have added the standard deviation in Fig 4, but the error bars are barely visible.

Comment 24: Please add a comparative discussion section.

Answer to comment 24: Discussion section was added, details are given in answer to comment 25.

Comment 25: The entire result explanation seems ok, but the results does not have any scientific discussion about comparisons.

Answer to comment 25: Some fragments have been moved from the results section, we have also added a comparison related to the effectiveness of removing other dyes. We also added the results of kinetics calculations and ANOVA. Therefore, in order not to unnecessarily extend the response for the Reviewer 3, we do not include them here. We kindly ask the Reviewer 3 to check directly in the corrected version of the article.

Comment 26: Conclusion and Future perspectives should be revised carefully. Conclusion section is missing some perspective related to the future research work, quantify main

research findings, and highlight relevance of the work with respect to the field aspect.

Answer to comment 26: Conclusion and future perspectives were revised carefully. Following statements were added: This research is the first stage of work to explore the possibility of using the UV/ZVI/H2O2 method for dye decomposition. In this stage, process parameters, including pH value and reagent doses for aqueous solutions of dyes, were determined. In the second stage, which is a continuation of the current work, treatment of real wastewater containing dyes is planned.

Comment 27: To avoid grammar and linguistic mistakes, MAJOR level English language should be thoroughly checked. Please revise your paper accordingly since several language

issue occurs on several spots in the paper.

Answer to comment 27: Thank you for pointing out the language defects in our manuscript. English is not our native language. In order to ensure the appropriate linguistic quality of the article, it has been submitted to the MDPI English Editing Service for linguistic corrections. We attach appropriate certificate.

Comment 28: Reference formatting need carefully revision. All must be consistent in one format. Please follow the journal guidelines.

Answer to comment 28: As suggested by the Reviewer 3, the references have been revised. Indicated articles were added, undesirable ones removed. Lumped references were removed. The references have been formatted in accordance with the requirements of the Materials journal.

Final comment: Decision = Major revision. It was tough for me to read and go through, but I was able to make a comment for improvement. Please put forth your best efforts and revised it. The idea is good so, I recommend a chance to revised it.

Answer to final comment: We have made every effort to ensure that the article is written in correct English after applying the corrections suggested by the Reviewer 3. Thank you again for giving us a chance to correct this article, comments and suggestions.

Round 2

Author Response

Dear Reviewer,

We would like to thank the Reviewer for the critical reading, as well as for helpful, relevant, and constructive remarks. The manuscript was corrected and rewritten in accordance with the suggestions, which improved the quality of our paper. All changes are indicated in red. We acknowledge that these modifications definitely improve the quality of our manuscript. We hope that the changes and explanations are acceptable and satisfactory with the expectation of the Reviewer.

You can find below the details of the modifications and explanations. Thank you very much for revising our manuscript again!

General comment: The revised manuscript addresses to a large degree the issues that were raised for the original version of the manuscript. Below are a few minor issues the authors may consider revising. Overall, the manuscript is OK to be considered for publication.

Answer to general comment: When preparing the previous version of our manuscript, we tried our best to respond to the Reviewer's comments and suggestions. We will make every effort to dispel all doubts of the Reviewer and clearly explain any inaccuracies.

Comment 1: The specific surface area and porosity studies seem not to contribute to any further discussion about how the ZVI works to decompose the dyes when UV radiation hydrogen peroxide are present. If there is little impact from the specific surface area and porosity, inclusion of these data in the main text may not be justified. They could be moved to supplementary materials.

Answer to comment 1: We agree with the Reviewer that specific surface area and porosity studies are of a minor importance. The text summarizing the results obtained is short, a few lines, so we ask the Reviewer for permission to leave it in the main body of the article, but we fully agree that the confirmatory results presented in the Table 1 and Figure 2 as taking up a lot of space should be moved to the supplementary material. They were moved to supplementary data, as Reviewer suggest. Former Table 1 is now Table S1, former Figure 2 is now Figure S1.

Comment 2: Line 264, ZVI samples have smaller surface areas rather than “larger”, as claimed by the authors.

Answer to comment 2: Word “larger” was changed into “smaller”.

Comment 3: Based on Figure 6, the pH studies show the AM E123 has the best stability between pH 4 and 6, and AM ACID is at pH 5. However, all the other properties were conducted at pH 3. Some clarification is needed for this.

Answer to comment 3: The Reviewer's observation is correct. The material is characterized by greater stability in conditions closer to neutral. This is, among other things, related to the solubility of individual forms of iron. However, in an acidic environment, iron will dissolve, Fe(II) ions will enter the solution, Fe(III) forms may and will appear, which is the result of the Fenton reaction. It is belived that optimal pH for Fenton process is 3.0. The aim of our research was to decompose the dye, which we considered more important than the stability of the catalytic material used. The process of dye decomposition itself results from the occurrence of radical reactions that are part of the Fenton process. The process can occur homogeneously or heterogeneously. In the case of a heterogeneous process, the stability of the catalyst is important, while in the case of a low stability of the catalyst, it degrades releasing iron ions into the solution, which initiate the Fenton process, which proceeds in a homogeneous manner. The subject of the difference and comparison between conducting the Fenton process classically homogeneously and heterogeneously with ZVI is quite well known scientifically. Our team also dealt with this, we have several publications related to it. Of course, a side effect of using pH 3.0 during process is the partial destruction of the catalyst and the need to separate Fe (II) ions from the treated wastewater - we did it by causing the final coagulation, raising the pH to 9.0 after a certain time of the process, which also allows the removal of unreacted hydrogen peroxide. In conclusion, we decided that it was worth risking damage to the catalyst in exchange for greater efficiency of the dye decomposition process. The catalyst is a readily available, cheap waste material anyway, so we did not consider its damage to be a significant problem. At the same time, we conducted all further analyzes for the effects of the treatment process at pH 3.0.

Comment 4: The title for Figure 8 needs to be more specific. There is no information to explain the difference among (a), (b) and (c) or among (d), (e) and (f).

Answer to comment 4: Due to the transfer to the supplementary material of the previous Figure 2, the numbers of the other Figures, has been changed. The Figure caption has been corrected. Currently figure caption is: Figure 7. Size characterization results for the catalysts after the process with numbers of the particles a), intensities b), and volumes c) of AM ACID and numbers of the particles d), intensities e), and volumes f) of AM E123.

Once again, thank you very much for your comments and suggestions.

Reviewer 3 Report

Accepted in the present form.

Author Response

We would like to thank the Reviewer for the critical reading, as well as for helpful, relevant, and constructive remarks.